

# Extending the range and reach of physically-based Greenland ice sheet sea-level projections

Heiko Goelzer[1], Constantijn J. Berends[2], Fredrik Boberg[3], Gael Durand[4], Tamsin Edwards[5], Xavier Fettweis[6], Fabien Gillet-Chaulet[4], Quentin Glaude[6,7], Philippe Huybrechts[8], Sébastien Le clec'h[8], Ruth Mottram[3], Brice Noel[6], Martin Olesen[3], Charlotte Rahlves[1,9], Jeremy Rohmer[10], Michiel van den Broeke[2], Roderik S.W. van de Wal[2,11]

[1] NORCE Research, Bjerknes Centre for Climate Research, Bergen, Norway
[2] Institute for Marine and Atmospheric research Utrecht, Utrecht University, Utrecht, the Netherlands
[3] Danish Meteorological Institute (DMI), Copenhagen, Denmark
[4] Univ. Grenoble Alpes, CNRS, IRD, Grenoble INP, IGE, 38000 Grenoble, France
[5] Department of Geography, King's College London, London, UK
[6] Laboratory of Climatology, Department of Geography, SPHERES research unit, University of Liège, Liège, Belgium
[7] Applied Computer Electronics Laboratory, University of Liège, Liège, Belgium
[8] Vrije Universiteit Brussel, Earth System Sciences and Departement Geografie, Pleinlaan 2, Brussel, Belgium
[9] Department of Earth Science, University of Bergen, Bjerknes Centre for Climate Research, Bergen, Norway
[10] BRGM, 3 av. C. Guillemin, 45060 Orléans, France
[11] Faculty of Geosciences, Department of Physical Geography, Utrecht University, Utrecht, the Netherlands

*Correspondence to*: Heiko Goelzer (heig@norceresearch.no)

**Abstract.** We present an ensemble of ice sheet model projections for the Greenland ice sheet (GrIS) that was produced as part of the European project PROTECT. The work makes use of ice sheet model (ISM) projections forced by high-resolution regional climate model (RCM) output and other climate model forcing, including a parameterisation for the retreat of marine-terminating outlet glaciers. The focus is on providing extended physically-based projections that improve our understanding of the range of GrIS future sea-level contributions and the inherent uncertainties over decadal to multi-centennial timescales. The experimental design builds on the Ice Sheet Model Intercomparison Project for CMIP6 (ISMIP6) protocol and extends it to more fully account for some of the uncertainties in sea-level projections. We include a wider range of CMIP6 climate model output, more climate change scenarios, several climate downscaling approaches, a wider range of sensitivity to ocean forcing and we extend projections schematically beyond the year 2100 up to year 2300, including idealised overshoot scenarios. GrIS sea-level rise contributions range from 16 to 353 mm in the year 2100 (relative to 2014), with strong dependency on the applied climate forcing. Contributions reach 49 to 3127 mm in 2300, indicative of large uncertainties and a potentially very large long-term response. We also extend the ISMIP6 forcing approach backwards over the historical period and successfully produce consistent simulations in both past and future for three of the four ISMs. The ensemble design of ISM experiments is geared towards the subsequent use of emulators to facilitate statistical interpretation of the results and produce probabilistic projections of the GrIS contribution to future sea-level rise.



## 1    Introduction

The Greenland ice sheet (GrIS) has transitioned from a near zero overall mass balance before the early 1990s to rapidly increasing mass loss that is ongoing today (van den Broeke et al., 2017). The driving mechanism of this change can be largely attributed to atmospheric and oceanic warming surrounding the ice sheet, which is amplified in the Arctic region compared to the global mean (Rantanen et al. 2022). This makes the ice sheet the currently largest single cryospheric contributor to global mean sea-level rise (e.g. Fox-Kemper et al. 2021). Projecting the future evolution of the GrIS is therefore an important element in providing sea-level practitioners with relevant information for adaptation planning and providing policy makers with guidance concerning the urgent need for mitigation, in line with the PROTECT project goals (Durand et al. 2022).

Projections of ice sheet contributions to future sea-level rise have recently been organised into a global community effort under the guidance of the Ice Sheet Model Intercomparison Project for CMIP6 (ISMIP6, https://climate-cryosphere.org/about-ismip6/). The initiative provided projections for both the Greenland and Antarctic ice sheets that served as the main source of the ice sheet sea-level projections (Fox-Kemper et al. 2021) in the latest report of the IPCC, AR6. The ISMIP6 GrIS projections (Goelzer et al., 2020a) used an experimental protocol (Nowicki et al., 2016, 2020) that included a regional climate model (RCM) to dynamically downscale global climate scenarios to the ice sheet scale. While only one RCM was used for the projections in ISMIP6 for feasibility reasons, recent work has revealed that different RCMs can show widely different behaviour under future climate change (Glaude et al., 2024). This strongly suggests that fully characterising uncertainties in ice sheet projections requires a broader sampling of climate forcing uncertainty, in particular pertaining to the downscaling process. In addition, the first wave of ISMIP6 projections was forced by output from CMIP5 models (Goelzer et al., 2020a) and a subsequent update used a small subset of the then available CMIP6 models forcing (Payne et al. 2021), both under only two scenarios (RCP8.5/SSP5-8.5 and RCP2.6/SSP1-2.6). Extending the forcing to a broader range of downscaled CMIP6 climate forcing and scenarios was therefore another major concern.

The ISMIP6 projections started with year 2015 and the protocol did not provide any guidance on how ice sheet modellers should initialise to that starting point. This led to a wide range of ice sheet histories preceding the GrIS projections, with numerous models not matching observed mass changes (Goelzer et al., 2020a; The IMBIE Team, 2020; Aschwanden et al., 2021). This also created challenges in presenting and combining projected ice sheet changes with observed changes in the AR6. Since then, it has become a priority to consider the historical experiment as part of the simulation and it has been shown that extending the ISMIP6 forcing protocol over the historical period can produce GrIS projections consistent with observed mass changes (Rahlves et al., 2025a).

While physically-based ice sheet model simulations like those produced by ISMIP6 now form an important basis of sea-level change projections, it requires statistical tools to generalise the results and make meaningful inferences. This need arises largely from the limited sampling of climate model forcing, model physics and parameter choices that remain relatively sparse due to practical limitations, computational cost and feasibility. A consequence is that some projections nowadays heavily rely on emulators (e.g. Edwards et al., 2021; Rohmer et al. 2022; Rohmer et al. 2025; Edwards et al., in prep) to help their



interpretation. Designing ice sheet experiments that can serve both direct interpretation of the result and feeding into emulators
has become an important consideration.

This paper presents a new set of physically-based GrIS sea-level projections designed to extend the ISMIP6 effort in several
aspects and to inform a next generation of emulator-based projections. We describe the experimental protocol, forcing and
models (Sect. 2), present results (Sect. 3) and close with a discussion (Sect. 4) and conclusions (Sect. 5).

## 73  2  Experimental setup

### 74  2.1  Experimental protocol

The ice sheet model experiments largely follow the ISMIP6 protocol for GrIS projections, which is documented in detail
elsewhere (Nowicki et al., 2016, 2020; Goelzer et al., 2018, 2020a). Here, we provide a brief summary of the main principles
and differences. Output from selected CMIP Earth system models (ESMs) serves as boundary conditions for i) RCMs
(producing the surface mass and energy balance and near-surface climate fields) and ii) for a retreat parameterization for
marine-terminating outlet glaciers (Slater et al., 2019; 2020). These, in turn, provide forcing for ice sheet models (Figure 1).





**Figure 1: General forcing approach for Greenland ice sheet model projections**

### 84  2.1.1  Forcing approach

The forcing from three RCMs, namely MAR (Delhasse et al., 2020), HIRHAM (Mottram et al., 2017) and RACMO (Noël et
al., 2018) and from a statistical downscaling approach (Noël et al., 2022), is provided in the form of annual surface mass
balance (SMB) and surface temperature (ST) anomalies relative to the period 1960-1989 (red box in Figure 2). In addition, we



provide estimates of local annual vertical SMB and ST gradients, that are used to propagate dynamic ice sheet elevation
changes when updating SMB and ST. The SMB forcing applied in the ice sheet model at a given time t is:
SMB(x,y,t) = SMBref(x,y) + aSMB(x,y,t) + dSMBdz(x,y,t) * dz(t),
Where SMBref [mm/yr] is the surface mass balance used by each individual ice sheet model during initialisation, aSMB
[mm/yr] is the SMB anomaly, dSMBdz [mm/yr/m] is the vertical gradient of SMB and dz [m] is the elevation change since
the start of the experiment. A similar approach applies to ST that can be used as boundary condition for the evolution of ice
temperature in the model. Differences in SMB/ST stem from the various ESMs and scenarios used to force the three RCMs.
The retreat of marine-terminating outlet glaciers is parameterised as an empirically derived function of ocean thermal forcing
(from the ESMs) and runoff (from the RCMs), both identified as the main drivers for melting of marine-based calving fronts
(green box in Figure 2, Slater et al., 2019; 2020). This is a relatively crude approximation for the complex interaction between
glaciers and the ocean that is poorly understood and difficult to resolve in large-scale ice sheet models. The uncertainty in this
forcing is captured by a retreat parameter *kappa*, that can be expressed probabilistically and is sampled at its median (med),
25th (high) and 75th (low) percentile value (like in ISMIP6) and in extension at its 5th and 95th percentile value. In some cases,
we have added control experiments without any prescribed retreat that are labelled 'pno'.

**CMIP6 models**
- selected based on availability
- quality check over historical period

**CMIP5 models**
- performance over historical period
- to maximise spread in projections

**Surface mass balance forcing**
- RCMs forced by CMIP models
- SMB anomalies relative to 1960-1989
- SMB elevation feedback parameterized based on d(Runoff)/dz

**Outlet glacier forcing**
- Prescribed outlet glacier retreat
- Empirically derived based on historical glacier retreat, ocean temperature and runoff
- κ parameter for sensitivity

**Ice sheet models**
- From PROTECT project partners


**Figure 2: Surface mass balance and retreat forcing**
**2.2      Regional climate model forcing**
The emphasis of this project is to extend the range of available forcings to a larger number of CMIP6 ESMs, scenarios (SSP1-
2.6, SSP2-4.5, SSP5-8.5) and to provide surface mass balance forcing from several RCMs: MAR (Delhasse et al., 2020),
RACMO (Noël et al., 2018) and HIRHAM (Mottram et al., 2017). In addition, we use forcing produced by a statistical
downscaling approach (SDBN1, Noël et al., 2016, 2020, 2022), which has been used here to translate ESM forcing from
CESM2-WACCM directly to the ice sheet scale. In the following, we will consider this approach included when using the



term RCMs. An overview of available forcing data is given in Table 1. Corresponding retreat mask forcing can be constructed
given sufficient output data from the RCMs and additional ESM ocean data, typically retrieved from the CMIP archives
(https://esgf-node.llnl.gov/projects/cmip6/). Forcing with MAR version 3.9 was produced for ISMIP6 and remained available
for ice sheet simulations under PROTECT.

**Table 1. SMB forcing data available for ice sheet modellers. MARv3.9 output was produced for ISMIP6.**

| CMIP | ESM | SSP1-2.6 RCP2.6 | SSP2-4.5 | SSP5-8.5 RCP5.8 |
|---|---|---|---|---|
| CMIP6 | CESM2 | MARv3.12 RACMO2.3p2 | MARv3.12 RACMO2.3p2 | MARv3.12 |
| | CESM2-Leo[†] | | | MARv3.9 MARv3.12 RACMO2.3p2 HIRHAM5 |
| | CESM2-WACCM | SDBN1[‡] | | SDBN1[3] |
| | CNRM-CM6-1 | | | MARv3.9 MARv3.12 |
| | CNRM-ESM2-1 | | | MARv3.9 MARv3.12 |
| | EC-Earth3 | HIRHAM5 | | HIRHAM5 |
| | IPSL-CM6A-LR | | | MARv3.12 |
| | MPI-ESM1 | MARv3.12 | MARv3.12 | MARv3.12 |
| | NorESM2-MM | | MARv3.12 | MARv3.12 |
| | UKESM1-0-LL | | MARv3.12 | MARv3.12 |
| | UKESM1-0-LL-Robin[†] | | | MARv3.12 |
| CMIP5 | ACCESS1.3 | | | MARv3.9 MARv3.12 |
| | CSIRO-Mk3.6.0 | | | MARv3.9 |
| | HadGEM2-ES | | | MARv3.9 |
| | IPSL-CM5-MR | | | MARv3.9 |
| | MIROC5 | MARv3.9 | | MARv3.9 |
| | NorESM1-M | | | MARv3.9 |

[†] pre-CMIP6 ensemble member. [‡]Direct statistical downscaling of CESM2-WACCM (Noël et al., 2022).



### 2.2.1 Required RCM data

The data required to produce ice sheet model forcing were developed during ISMIP6 in collaboration with the developers of
MAR, the only RCM used to generate projections for the project at the time. This includes extension of the RCM forcing
beyond the observed ice sheet mask and producing output needed for vertical adjustment of the forcing to a changing ice sheet
topography. In MAR this is done with the same statistical downscaling method used to produce results at 1 km resolution
(Franco et al., 2012) as done in the GrSMBMIP intercomparison (Fettweis et al., 2020). In RACMO and SDBN1, vertical
gradients were estimated following Noël et al. (2016) combining statistically downscaled SMB components with surface
elevation and ice mask from the GIMP DEM (Howat et al., 2014), down-sampled to 1 km spatial resolution. Vertical gradients
were first computed on ice-covered grid-cells using SMB components and surface elevation of the current grid-cell and at least
five (up to eight) neighbours and further extrapolated outside the ice sheet to cover the tundra region.
In HIRHAM5, gradients are produced at a 5 km horizontal resolution using an updated subsurface scheme (Langen et al.,
2017). These gradients are subsequently bilinearly interpolated to the 1 km MAR grid. Outside the observed ice mask,
extrapolation to cover the tundra is performed via distance-weighted averaging, followed by smoothing using weighted
averages of the grid points, including the eight surrounding points.

To facilitate use of RCMs and other downscaled climate forcing in PROTECT and other projects, we outline a detailed data
request in Appendix 1.

### 2.3 Forcing dataset preparation

Output from RCMs and ESMs is collected and processed using methods established during ISMIP6. The aim is to provide a
consistent forcing dataset for ice sheet modellers in familiar format. It requires interpolation of RCM output to a common grid
at 1 km resolution, calculating anomalies and adjusting units and file formats. Retreat mask forcing is produced based on the
initial ice sheet mask for each individual participating ice sheet model and version. All data are provided in NetCDF format
following the ISMIP6 guidelines (https://theghub.org/groups/ismip6/wiki/ISMIP6-Projections-Greenland).

### 2.4 Participating ice sheet models

The ensemble includes four numerical ice sheet models that are routinely run by the participating partners for GrIS simulations
(IMAU, VUB, IGE, NORCE). A brief overview of the model characteristics is given in Table 2 and short model descriptions
are given below.

**Table 2. Ice sheet model names and characteristics. SIA - Shallow ice approximation to the force balance, SSA - Shallow shelf**
**approximation, HO – higher order approximation (Fürst et al. 2013), DIVA - variationally derived, depth-integrated approximation**
**(Goldberg 2011).**

| Group-Model | Type | Resolutions (km) | Variants |
|---|---|---|---|



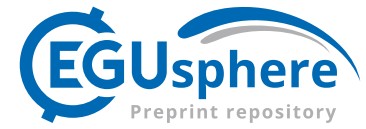

| IMAU-IMAUICE | SIA-SSA, regular grid | 10, 16, 20, 30, 40 | Sliding law, spinup |
| VUB-GISM | Regular grid | 5 | HO, SIA |
| IGE-ELMER | SSA, finite element | 1 – 6 (variable) | Sliding law |
| NORCE-CISM | DIVA, regular grid | 2, 4, 8, 16 | Initial and historical SMB |


### 2.4.1 IMAU-IMAUICE

The model (Berends et al., 2022) is initialised using a hybrid approach, combining a basal inversion method (Berends et al.,
2023) with a paleoclimate spin-up. During the inversion phase of the initialisation, spatial patterns in basal slipperiness are
iteratively adjusted until the modelled ice sheet reaches a stable state that closely matches the observed present-day ice sheet
geometry (Morlighem et al., 2017) and surface velocity (doi: 10.24381/cds.0b96b838). The prescribed climate is fixed at
present-day conditions: monthly mean values of 2-m air temperature and total precipitation, which are obtained from the 1950-
1980 mean of the ERA5 reanalysis (Hersbach et al., 2020). The SMB is calculated from these quantities using the IMAU-ITM
model, which is calibrated to RACMO2.3p2 over the 1979-2014 period (Fettweis et al., 2020). The steady-state geometry and
basal slipperiness resulting from the inversion phase are then used to initialise the model during the last interglacial, 120,000
years ago. The climate evolution of the last glacial cycle is then prescribed using a matrix method (Berends et al., 2018), based
on different pre-calculated GCM output for the different IMAU-ICE versions: either HadCM3 (Singarayer and Valdes, 2010),
CCSM (Brady et al., 2013), or the PMIP3 best-performing ensemble mean (Scherrenberg et al., 2023). Climate evolution
during the historical period is approximated by forcing the climate matrix with the Law Dome ice-core $CO_2$ record
(MacFarling Meure et al., 2006), subjected to a 60-year smoothing representing the delayed response of the climate to changes
in $CO_2$.

### 2.4.2 VUB-GISM

VUB-GISM (Huybrechts, 2002; Fürst et al., 2013; 2015) is configured either with the higher order or a shallow ice
approximation to the force balance. GISM was initialised to the present-day geometry by assimilation of the observed ice
thickness (Le clec'h et al., 2019). A steady state was assumed for the starting date of December 1989 using the 1960–1989
mean SMB from MAR forced by the ERA5 meteorological reanalysis climate. The iterative initialization method optimised
both the basal sliding coefficient in unfrozen areas and the rate factor in Glen's flow law for frozen areas. The ice temperature
and the initial velocity field needed in the initialization procedure were derived from a glacial spin-up with a freely evolving
geometry over the last two glacial cycles with a synthesised temperature record based on ice-core data from Dome C, NGRIP,
GRIP and GISP2 (Fürst et al., 2015). For this spin-up experiment, a PDD model was used with an observed precipitation field
derived from the Bales et al. (2009) surface accumulation for the period 1950–2000 and scaled by 5 % °C$^{-1}$. The ice
temperature and velocity fields from the "free geometry present-day" were rescaled to the observed ice thickness (Morlighem
et al., 2017) and excluded peripheral ice (Citterio et al., 2013). The historical experiment is run from January 1990 to December



2014 using the yearly SMB from MAR forced by ERA5 meteorological reanalysis. For the projections, the standard retreat
forcing from the ISMIP6 protocol is applied.

### 2.4.3    IGE-ELMER

The model is initialised using an inverse control method as in Gillet-Chaulet et al. (2012) to calibrate the basal friction
coefficient field. For the momentum equations, we solve the shelfy-stream approximation with a sub-grid parameterization of
the friction for partially grounded elements. The vertically averaged viscosity is constant in all simulations and is initialised
using the temperature field coming from a palaeo-spin-up (125 kyr) of the SICOPOLIS model. The basal friction coefficient
is constant in all transient simulations and is initialised with the control method so that the mismatch between observed and
modelled surface velocities is minimum. As observations, we use a composite from the NASA Making Earth System Data
Records for Use in Research Environments (MEaSUREs) Greenland Ice Sheet Velocity Map (V1) (Joughin et al., 2010). The
ice sheet topography is initialised using the IceBridge BedMachine Greenland V3 data set (Morlighem et al., 2017). The ice
sheet model is then relaxed for 20 years using a constant surface mass balance given by the 1960–1989 mean SMB from the
regional climate model MAR v3.12 forced with ERA5 (Fettweis et al., 2017). The calving front positions are fixed during the
relaxation. We use an anisotropic mesh with a horizontal resolution ranging for 1 to 6 km. For the projections, the standard
ISMIP6 protocol is applied and we test the sensitivity to different friction laws: a linear friction law, Weertman friction law
with m=1/3 and a parameterised Coulomb friction law.

### 2.4.4    NORCE-CISM

The Community Ice Sheet Model (CISM; Lipscomb et al., 2019) is run using a depth-integrated higher-order velocity solver
based on Goldberg (2011) and a basal-sliding law based on Schoof (2005). The ice sheet is initialised with present-day
thickness and bed topography (Morlighem et al., 2017) and an idealised temperature profile. CISM is then spun up for 5 000
years with surface mass balance and surface temperature from a 1960–1989 climatology provided by the MAR regional climate
model (Fettweis et al., 2017) and with basal heat fluxes from Shapiro and Ritzwoller (2004). During the spin-up, the model is
nudged toward present-day thickness by adjusting friction coefficients in a basal-sliding power law. There is no dependence
of basal sliding on basal temperature or water pressure. All floating ice is assumed to calve immediately. For partly grounded
cells at the marine margin, basal shear stress is weighted using a grounding-line parameterization. By the end of the spin-up,
the ice thickness, temperature and velocity fields are very close to steady-state and closely match the provided observed
geometry and also the observed horizontal velocity, which is not used during initialisation. For the historical period (1960–
2014), the model is run forward with SMB and surface temperature anomalies, including lapse-rate corrections, from the MAR
simulation that provided the background climatology and with retreat forcing of various sensitivities. Basal friction coefficients
are held fixed at the values obtained during the spin-up. The different CISM model versions used here differ by the horizontal
grid resolution (2 - 16 km), by the RCM version used for spinup and historical run (MARv3.9 vs MARv3.12) and by the
sensitivity of the retreat parameterisation applied over the historical period.






## 2.5 Experiments

### 2.5.1 Ice sheet initialisation and historical run

Under ISMIP6 protocol, ice sheet modellers were free to initialise their model as they wish, with the aim to produce a present-day state of the ice sheet that is close to observations. This procedure may involve a historical experiment that brings the ice sheet into a state that is assigned to the end of 2014. In contrast to this freedom in setting up the model, the projections 2015-2100 that then follow are very tightly constrained by the forcing. This is also the case for the retreat forcing, which takes the individual 2014 ice mask as a reference and provides masks that impose the position of the (retreated) calving fronts forward in time. For PROTECT we have extended this approach by providing retreat forcing before 2015 that is calculated from reconstructions of past runoff and ocean thermal forcing (see Rahlves et al., 2025a). This allows for a consistent forcing of the models in past and future and considers historical retreat of the outlet glaciers, which was an important source of mass loss after 1990 (The IMBIE Team, 2020). We can now interpret the experiment leading up to 2015 as a real historical simulation. The ISMIP6 practice of removing the results of an unforced control experiment from the projections is therefore not needed here.

The practice of including the historical experiment as part of the experimental design (which was not the case for ISMIP6) should ultimately imply that any variation in the ISM modelling choices should be represented in this experiment. As a consequence, each model variant would in principle require a separate historical experiment, so that modelling choices remain consistent at the beginning of the projections (here in year 2015). At the beginning of the project, we did not consider this constraint and most ice sheet model runs were conducted with a single historical experiment (like in ISMIP6) with medium retreat sensitivity, which then branches into projections with different sensitivity. We later conducted some experiments with consistent retreat forcing sensitivity with NORCE-CISM.

### 2.5.2 Future projections to year 2100

The future projections from 2015 to 2100 follow the ISMIP6 forcing protocol with SMB anomalies and retreat forcing applied as described in Sec. 2.1.1. With the available forcing described in Sec. 2.3, we obtained output from 14 different global models, forced with three different scenarios (SSP1-2.6, SSP2-4.5, SSP5-8.5) and downscaled with three RCMs and one statistical downscaling method.

### 2.5.3 Extensions after year 2100

Few CMIP6 models have carried out the scenarioMIP extensions (O'Neill et al., 2021) to 2300, and even fewer have provided 6-hourly output typically required for RCMs to downscale the data. We have currently only three examples of ice sheet forcing with what we will refer to as 'natural extensions' beyond 2100 from the ESMs IPSL-CM6A-LR (for scenarioMIP SSP5-8.5-



ext) and CESM2-WACCM (for scenarioMIP SSP5-8.5-ext and scenarioMIP SSP1-2.6-ext). CESM2-WACCM has been
statistically downscaled with SDBN1 (not requiring 6-hourly output) and IPSL-CM6A-LR has been dynamically downscaled
with variants of MARv3.12. There is one extension to 2200 with MARv3.12 downscaling IPSL-CM6A-LR under scenario
SSP5-8.5/SSP5-8.5-ext using the same approach as for the other experiments. The MAR modellers questioned the validity to
continue downscaling the relatively strong climate forcing from IPSL-CM6A-LR SSP5-8.5-ext at a fixed present-day
topography beyond 2200, given that the ice sheet geometry should have considerably changed by then, hence impacting SMB
(Delhasse et al., 2024). We have therefore performed two additional pilot experiments with different topography updates
extending to 2300 (MARv3.13-e05 and MARv3.13-e55). The construction of these forcings is described in more detail in
Appendix 3. The retreat mask forcing can in principle be constructed in the same way as for the experiments extending to
2100. However, the underlying assumptions of the parameterisation may not hold for the very large retreat distances produced
under sustained very strong warming to 2300. Because of that we have already limited the retreat sensitivity to the 25-75
percentile range for the natural extensions, but caution that these simulations show higher uncertainty.
In addition to the natural extensions, we have designed schematic extensions of the forcing data to the year 2300 to evaluate
the longer-term response of the ice sheet for a broader range of ESMs. The first set of extensions is carried out by repeating
the forcing of the last ten available years (2091-2100) in randomised order and keeping the retreat mask of year 2100 constant.
Aside from the obvious shortcoming that this is a schematic extension, another problem on this timescale may be that the
climate response to changing ice sheet geometry is not properly accounted for. Furthermore, the formulation of the retreat
forcing implies a constant mask for stabilising the forcing, which may underestimate the retreat. Alternative prolongations
could be envisioned, thus the current approach should be considered a pilot experiment and not a guide to produce realistic
scenarios.
For a second type of schematic extension, we have designed overshoot scenarios mimicking SSP5-3.4-OS by reusing the
regular SSP5-8.5 forcing before 2100 and simulating a climate cooling and corresponding increase of the SMB until 2300.
These overshoot scenarios are constructed using global mean temperature as a proxy for the temperature and SMB evolution
by sampling existing yearly forcing data until 2055 and reorganising them to new time series until 2300. The shape of the
global temperature proxy evolution is parameterised and has been calibrated to a few existing ESM results (CESM2-WACCM,
IPSL-CM6A-LR, MRI-ESM2-0) for overshoot scenario SSP5-3.4-OS. The resulting time series are illustrated in Appendix 3.
**2.6      Data request for ice sheet model output**
The requested ice sheet model data consists of the most important diagnostic output at annual time resolution, such as ice
thickness, bedrock and surface topography, horizontal velocities and integral mass balance terms. We are following the ISMIP6
data request format (https://theghub.org/groups/ismip6/wiki/ISMIP6-Projections-Greenland).



**2.7      Ensemble design**
The collection of forcing data covers a wide range of variations across different ESMs and greenhouse gas (GHG) scenarios,
but ultimately represents an 'ensemble of opportunity'. This is even more true for the selection of RCMs and ISMs, which is
limited to available models in the consortium. PROTECT has therefore conceptualised and operated from the beginning a
modelling strategy that embeds the physically-based modelling into a wider framework allowing for a statistically meaningful
probabilistic interpretation of the results.
In order to facilitate the sampling strategy in that framework, experiments in the ensemble are labelled by 6 characteristics that
are colour-coded in Figure 3 and given in square brackets in the following. Setting up a specific ice sheet simulation requires
climate forcing (SMB and ST) for a given choice [Global model, GHG scenario, Regional model] and retreat mask forcing for
a given choice [Global model, GHG scenario, Regional model, Retreat sensitivity]. We furthermore have different ice sheet
models built on a certain [Code base] (here referred to by the ISM name) and they are operated using certain modelling
[Choices] (initialization strategy, approximations, parameterizations, parameter choices). In our current approach, different
sets of modelling choices are summarised and assigned to a specific model version number. However, the impact of specific
modelling choices could be further analysed e.g., by using the technique described by Rohmer et al. (2022; 2025).

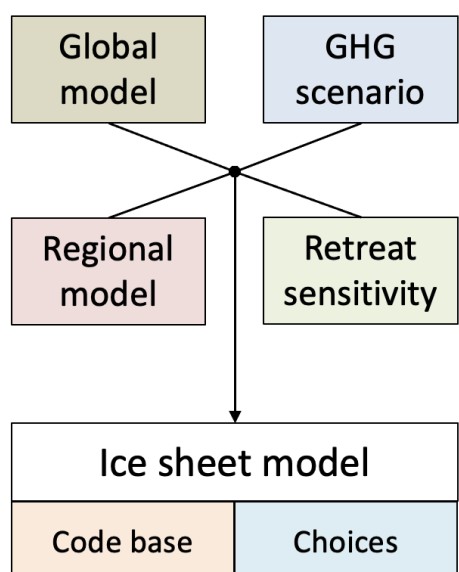


**Figure 3. Forcing and model options relevant for the larger ensemble design.**

The current set of results discussed below is a broad sampling of the available forcings and parameter choices to cover a wide
range of possible projections and their uncertainties. Based on feedback from the researchers running emulators using these
results, we have iteratively updated the ensemble with additional simulations to refine the sampling for specific choices where



needed. The repeated extensions and overshoot scenarios are examples of additional experiments that were deemed important
to improve the emulator performance for predictions up to 2300.

## 3       Results

The following results are presented as an overview of available ISM simulations and provide insight into the typical ranges
and main uncertainties. We have produced 1472 individual ice sheet model projections that form the ensemble of GrIS results.
An overview of the used ISM model versions is given in Table A2. Sea-level contributions are calculated taking into account
density (and bedrock adjustment for IMAUICE) following Goelzer et al. (2020b).
Figure 4 illustrates the typical time-dependence of the projections from the ensemble, with output from one model version per
group under the range of ESM and RCM forcing with median outlet glacier retreat sensitivity. It also shows historical
simulations of various lengths for the different ISMs. Under this forcing, which includes scenarios SSP1-2.6, SSP2-4.5 and
SSP5-8.5 for various ESMs, all sea-level contributions are increasing and positive by the year 2100. Judging by average mass
loss rates over the last 30 years, none of the simulations shows signs of ice sheet stabilisation (zero or positive mass change)
towards the end of the experiment, but rather continued mass loss, suggesting larger to much larger contributions for time
scales beyond 2100.

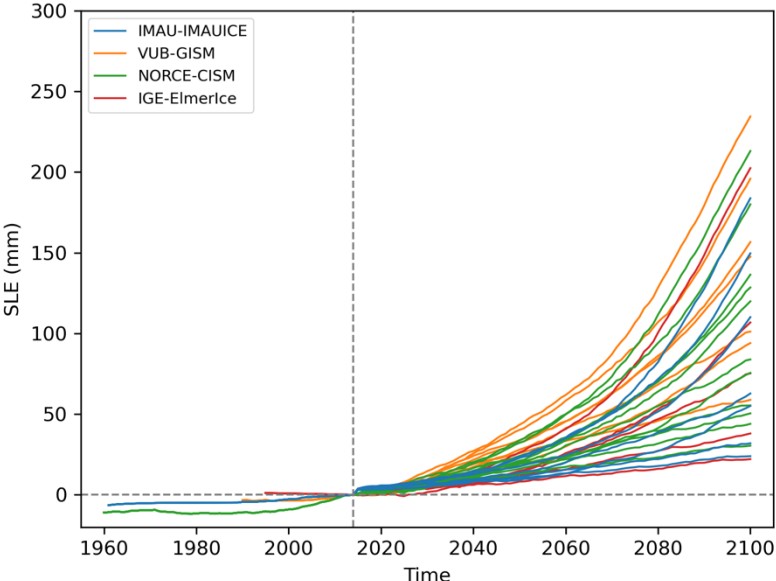


**Figure 4. Projected sea-level contribution from the GrIS until 2100 from the four participating ice sheet models (one model version per group), median retreat sensitivity and forcings produced specifically for PROTECT (MARv3.12, RACMO2.3p2, HIRHAM5). The aim of this figure is to illustrate the range and distribution of the projections, not individual members.**





Results for the year 2100 of the whole ensemble of projections with all available scenarios, ESMs, RCMs and ISMs under five
different retreat sensitivities (5th percentile, high, med, low, 95th percentile) are summarised in Figure 5. We have not included
results for the experiments that continue to 2200 and 2300 here, which are instead shown in Figure 8 with results at the
respective ends of the simulations. The contributions in the year 2100 (relative to 2014) lie in a range between 16 and 353 mm,
with the largest numbers from experiments that combine high climate sensitivity (UKESM1-0-LL variants) and very high
retreat sensitivity (5th percentile).

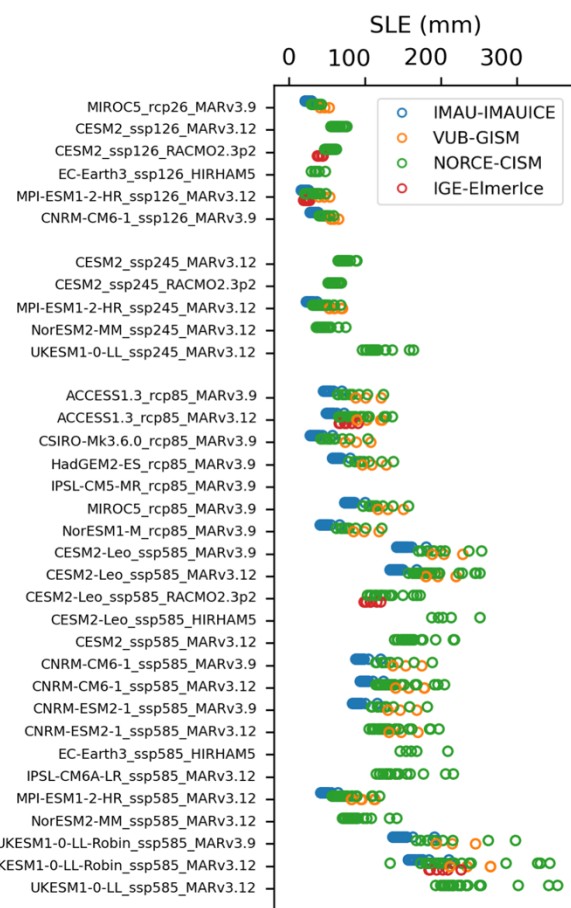


**Figure 5. Overview of produced GrIS sea-level projections for the year 2100 from 4 ice sheet models (23 different model versions) and 5 retreat sensitivities (med, high, low, p95, p05).**


Figure 6 illustrates ISM results of the runs to 2100 sorted by different categories. The comparison between ISMs (a), RCMs
(b), scenarios (c) and CMIP iterations (d) shows primarily the sampling frequency across the ensemble. Unequal sampling
limits the direct interpretation of the results, but some conclusions can be drawn, nevertheless. The range of results for the
different ISMs is largely similar (Figure 6a) and only larger for CISM because a wider range of retreat parameters (5th - 95th



percentile range) was sampled with this model. Simulations forced with regional models MAR and HIRHAM5 (Figure 6b)
show higher contributions under high climate forcing compared to RACMO, which is in line with SMB results discussed
recently (Glaude et al, 2024). The scenario ranges (Figure 6c) of projected sea-level contributions from the GrIS by the year
2100 (relative to year 2014) are 16-76 mm (SSP1-2.6/RCP2.6), 22-163 mm (SSP2-4.5) and 27-353 mm (SSP5-8.5/RCP8.5).
These results indicate a very large range of sea-level contributions in particular under forcing scenario RCP8.5/SSP5-8.5.
Figure 6d shows an increased sensitivity from CMIP5 to CMIP6, confirming earlier results (e.g. Hofer et al., 2020; Payne et
al., 2021), although unequal sampling is an additional factor.

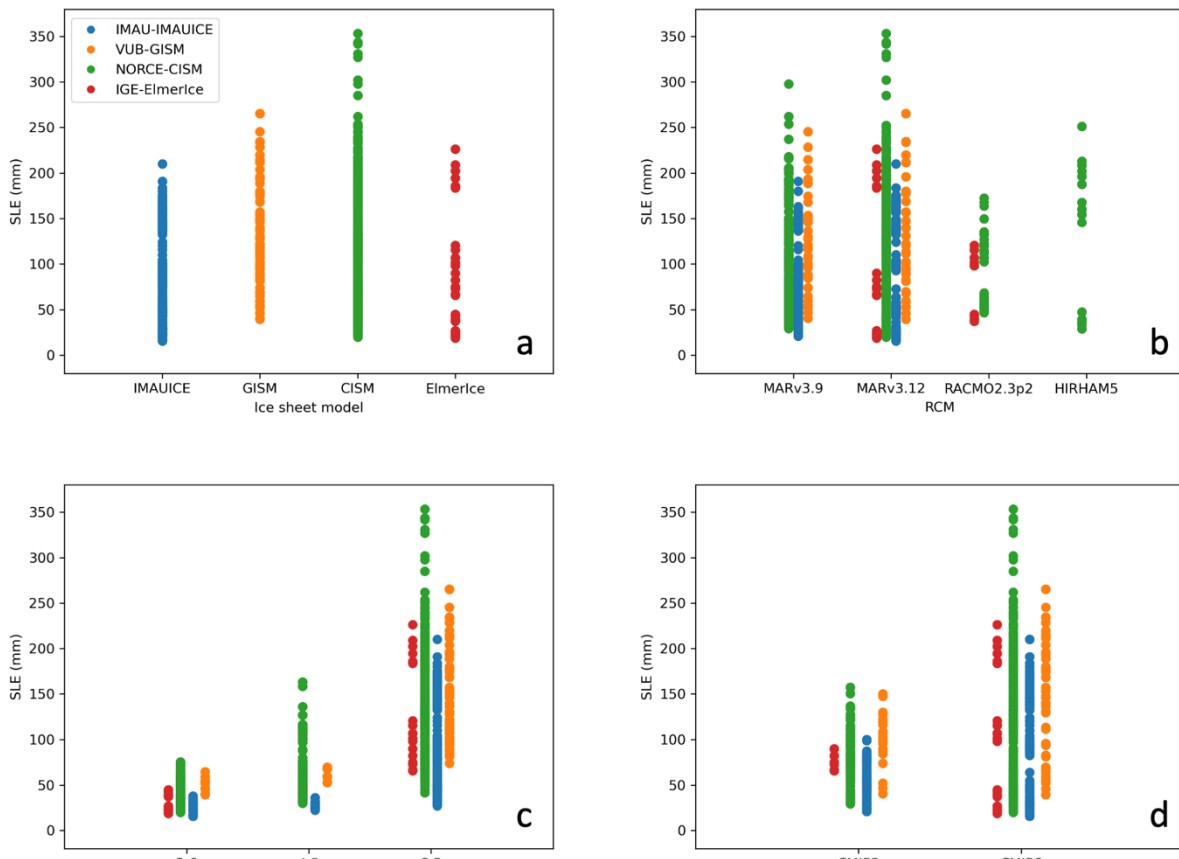

**Figure 6. Sea-level contribution from the GrIS sorted by a) ice sheet model, b) regional climate model, c) scenario and d) CMIP**
**ensemble. The colour legend is the same for all panels. Scenarios labelled '2.6' in c) include SSP1-2.6 and RCP2.6 and '8.5' includes**
**SSP5-8.5 and RCP8.5.**

Uncertainty in the projections arises from the climate forcing (different ESMs, scenarios), the translation of the forcing to the
ice sheet scale (RCMs/downscaling, retreat parameterisation) and from the ISMs themselves. We have quantified these



uncertainty ranges by comparing experiments with one of the factors modified at a time and averaging over available subsets.
Under SSP5-8.5/RCP8.5 forcing, the ESM choice explains a range of 130 mm (cf. Figure 6c), compared to a range of 84 mm
for RCMs (cf. Figure 6b), 50 mm for ISMs (cf. Figure 6a) and 13 mm for retreat forcing (25th - 75th percentile range).
Figure 7a shows results for a schematic prolongation to 2300 for one of the ice sheet models with repeated SMB forcing and
constant retreat mask after year 2100. It illustrates that sea-level contributions from Greenland continue to increase well beyond
year 2100 even under stabilised forcing. Contributions can exceed 1.2 m (under very high retreat forcing) by 2300 for
prolonged SSP5-8.5/RCP8.5 but may stabilise for prolonged SSP1-2.6/RCP2.6 somewhere below 200 mm. Results from the
schematic overshoot scenarios, mimicking SSP5-3.4-OS (Figure 7b) shows stabilisation for three out of the nine experiments
(CESM2-WACCM SDBN1, CESM2-Leo RACMO2.3p2, NorESM2-MM MARv3.12), while the others have an ongoing near-
linear mass loss trend at the end of the experiments by 2300. The natural extensions to 2300 (Figure 7c) for CESM2-WACCM
SDBN1 show a range between 92 mm (SSP1-2.6) and 3127 mm (SSP5-8.5), indicating a strong dependence on the climate
forcing. large uncertainties and a potentially very large long-term response. Results for IPSL-CM6A-LR SSP5-8.5-ext show
that including a topography update (MARv3.13-e55) leads to a 6 % larger contribution in 2300 compared to calculating the
SMB for a fixed surface elevation (MARv3.13-e50). This is in addition to the parameterised SMB-height feedback active in
both experiments.





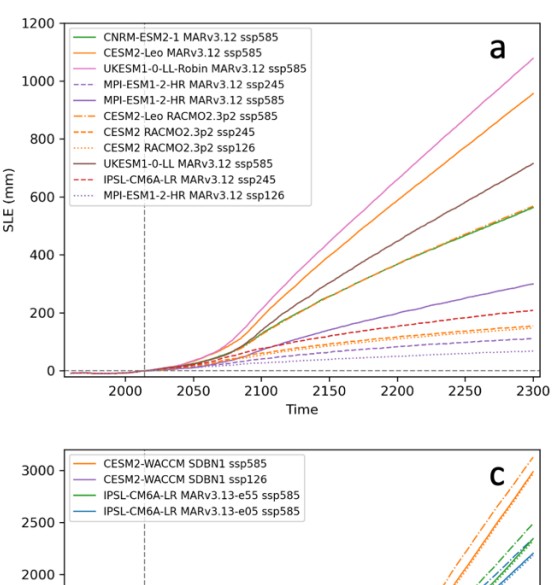

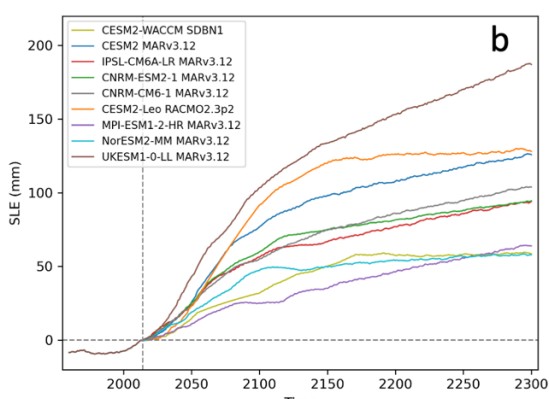

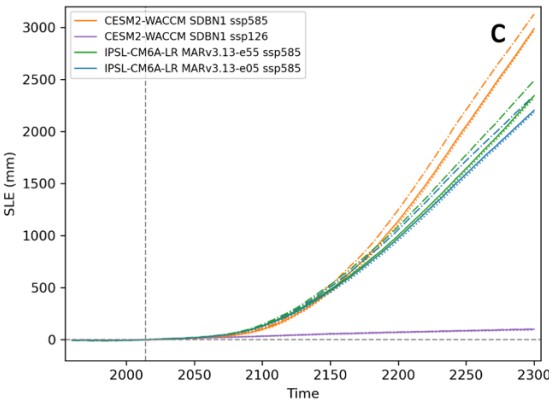

**Figure 7. Extensions to 2300 with NORCE-CISM for various ESMs/RCMs with median values of retreat parameter kappa for a) repeat forcing after 2100 and b) overshoot scenario mimicking SSP5-3.4-OS. c) Natural extensions with CMIP6 forcing until 2300 with median (solid), high(dot-dashed) and low (dotted) values of retreat parameter kappa. For scenario SSP1-2.6 experiments with various values for kappa are largely overlapping and difficult to distinguish. Extensions to 2200 are overlapping with the respective continuations to 2300 and are not shown.**

Figure 8 summarizes results at the end of the experiments for all schematic prolongations to 2300 (overshoot: o2300 and repeat: r2300) and also includes the natural extensions to 2200 and 2300 for CESM2-WACCM and IPSL-CM6A-LR. Note that results for the overshoots and lower scenarios on the left in Figure 8 are displayed on a different vertical scale compared to results under the high scenario on the right.

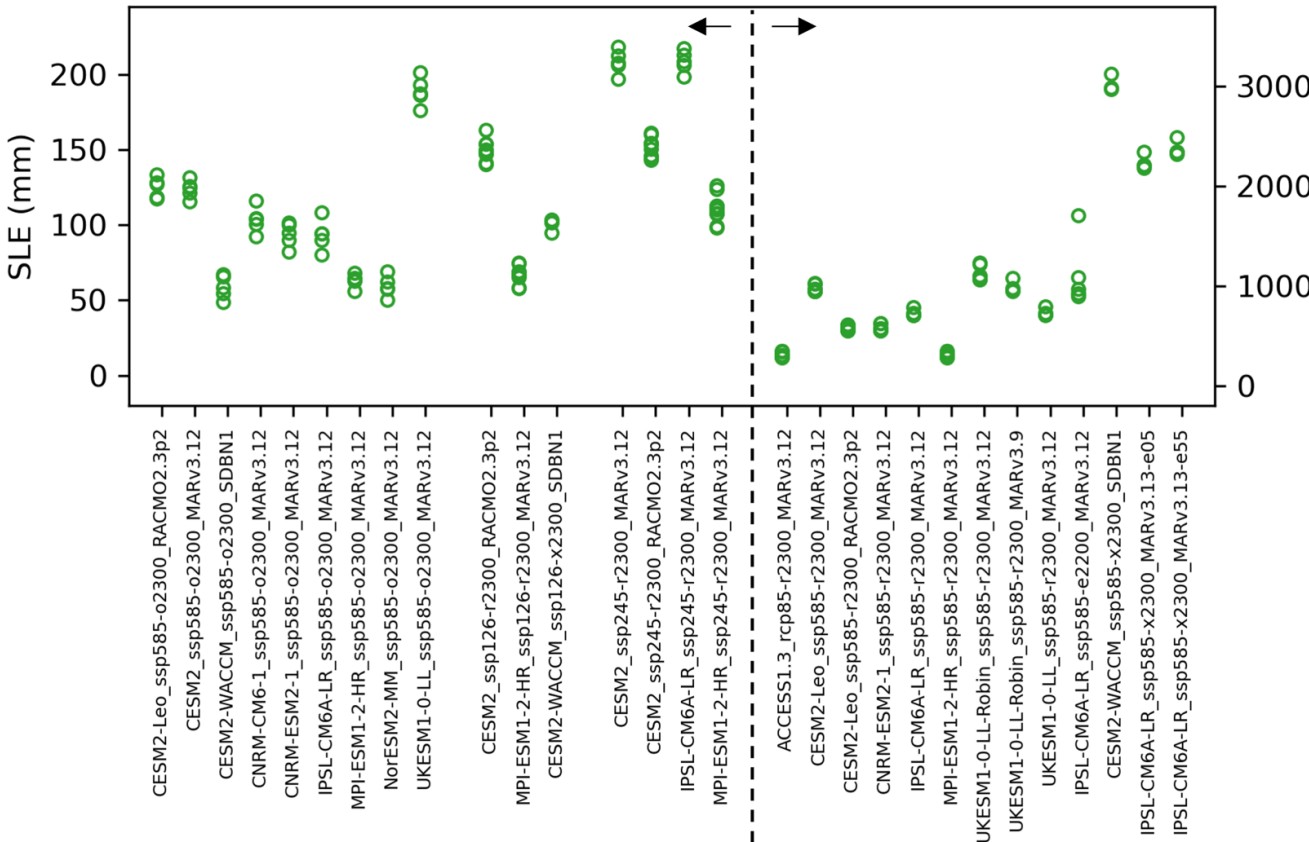

**Figure 8. Results for extensions to the year 2300 with NORCE-CISM for various ESMs/RCMs. o2300 - overshoot scenario, r2300 - repeat forcing, x2300 – regular ScenarioMIP extension, e2200 – regular ScenarioMIP extension to 2200. The leftmost experiment forced with MARv3.13-e55 has a topography update during the MAR simulation. Note different vertical scaling left and right from the dashed vertical line.**

## 4        Discussion

The range of projected sea-level contributions at 2100 largely overlaps with ISMIP6 results (Goelzer et al., 2020a; Payne et al., 2021) for the scenarios and forcings covered in both ensembles (SSP1-2.6/RCP2.6 and SSP5-8.5/RCP8.5). Slightly larger ranges here are due to additional ESMs, RCMs and, at the upper end, mainly due to the larger sampled range of the retreat parameter. We have added a solid number of experiments for the intermediate scenario SSP2-4.5, that was not represented in ISMIP6. Inclusion of these results does not increase the range but adds additional information for the subsequent emulation.



Results for experiments with the same climate model (CESM2-Leo) but different RCMs (MAR, RACMO, HIRHAM) mirror
the results from a comparison of the underlying SMB (Glaude et al., 2024), with considerable differences in the projected sea-
level contribution due to the choice of RCM. In addition, a larger relative contribution from experiments forced with HIRHAM
here compared to the SMB results (Glaude et al., 2024) is related to the way SMB is extended beyond the ice sheet mask and
how the vertical gradients are determined for parameterising the SMB-height feedback. In combination, this highlights the
urgent need to include uncertainty due to climate downscaling from global to ice sheet scale in the projections, which was
likely under-represented in ISMIP6 due to the use of only one RCM and only one method to take the SMB-height feedback
into account.
Uncertainties in the projections in this 'ensemble of opportunity' arise from sampling of ESMs, RCMs, ISMs and retreat
parameters, which implies that statistically meaningful interpretation of the raw model output is challenging. We have therefore
mostly limited the interpretation of results to typical ranges and leave finer-grained analysis to downstream efforts (e.g. Rohmer
et al., 2024, 2025; Edwards et al., 2021; 2025). Under the high forcing scenario SSP5-8.5/RCP8.5, global climate model
uncertainty (here choice of ESM) is dominating and explains a total range of 130 mm in the projections to 2100. This is
compared to a range of 84 mm for choice of RCM (not sampled in ISMIP6), and 50 mm for the choice of ISM, which is similar
to ISMIP6, despite the smaller number of ISMs in the present work. The range of 13 mm for retreat forcing (25[th] - 75[th] percentile
range) is slightly smaller compared to ISMIP6 (19 mm), but increases considerably to 52 mm when extending to the 5[th] - 95[th]
percentile range that we have explored here in addition.
Schematic extensions with repeated forcing to 2300 from a subset of ESMs (and only one ice sheet model) show an upper
range of contributions exceeding 1.2 m for prolonged SSP5-8.5/RCP8.5 and potentially stabilising contributions for prolonged
SSP1-2.6/RCP2.6 below 25 cm. These results are given with the caveat of the underlying schematic experimental setup and a
limited ensemble size. In comparison, regular ScenarioMIP extensions under scenario SSP5-8.5-ext that we have for two global
models (IPSL-CM6A-LR, CESM2-WACCM) produce contributions in the year 2300 exceeding 2.5 m and 3 m, respectively.
This is in strong contrast to results under CESM2-WACCM SSP1-2.6-ext with only 92 mm, underlining that the climate
scenario is the dominant source of uncertainty. On these timescales and under such forcing, feedbacks between ice sheet and
climate and how they are taken into account become first-order effects and introduce large uncertainties. In addition, the retreat
forcing approach has to be considered with caution for extended time periods in particular under high forcing scenarios.
Combined, these results indicate that standalone ice sheet simulations well beyond year 2100 likely require modifications to
the ISMIP6 forcing protocols and new methods to account for a changing ice sheet geometry (e.g. Goelzer et al., 2020c;
Delhasse et al., 2024; Rahlves et al., 2025b). Nevertheless, the experiments with repeated forcing give an approximate idea of
how stabilising forcing (and climate) at different levels could play out. On the considered timescale, stabilising the forcing has
the effect of stabilising the rate of change, not the ice sheet itself (unless the rate is close to zero). Results from the schematic
overshoot scenarios, mimicking SSP5-3.4-OS, were added specifically to provide the emulator with additional, complementary
information on ice sheet changes under forcing that does not follow a continuous increase in temperature. Results under this





forcing show that three (CESM2-WACCM SDBN1, CESM2-Leo RACMO2.3p2, NorESM2-MM MARv3.12) out of the nine
experiments with different climate model forcing produce what seems like a stabilising GrIS towards 2300.
**5        Conclusions**
We have produced a large ensemble of Greenland ice sheet projections with four different ice sheet models under various
forcings drawn from a wide range of ESMs, scenarios, RCMs, and retreat parameters. Uncertainty in the ice sheet models is
furthermore sampled with various model versions that differ by horizontal grid resolution, applied sliding law, and initial
states. This contribution to the European project PROTECT extends the projections of ISMIP6 in several important regards,
with an additional, intermediate scenario, several different RCMs, and more CMIP6 models. Results from different extensions
up to 2300 give a perspective on challenges for standalone simulations on this time scale.
**Appendix A: Data request for climate model output used as ice sheet forcing**
This section describes the climate model output required to construct ISMIP6-type ice sheet forcing for Greenland ice sheet
projections.
**Surface mass balance (SMB):** annual cumulative SMB [mm/yr w.e.]
Like most variables, the SMB needs to be extended outside of the observed ice sheet mask to accommodate ice sheet models
with a slightly larger than observed footprint. See main text how this was done in the different downscaling procedures.
**Vertical gradients of runoff:** annual mean slope of the local runoff-elevation gradients [mm/yr w.e. per m].
This variable is needed to parameterise the SMB-height feedback in ice sheet models. The gradients are expected to be
predominantly negative as runoff generally declines with elevation and should be masked to 0 where no runoff is present. This
variable has to be relatively smooth. Using gradients in runoff rather than gradients in SMB to parameterize the SMB-heigh
feedback is chosen because precipitation does not have consistent gradients with elevation. Extended outside of the observed
ice sheet margin.
**Skin temperature (Tskin):** annual mean skin temperature [degree C]
Used to force the thermodynamic ice sheet solution at the upper boundary. Extended outside of the observed ice sheet margin.
**Vertical gradients of Tskin:** annual mean **slope** of the local temperature-elevation gradient [degree C per m].
This variable is used to apply a lapse-rate correction of the temperature boundary condition with changing surface elevation.
This variable should be relatively smooth. Extended outside of the observed ice sheet margin.
**Runoff:** monthly cumulative runoff [mm/yr w.e.].
This variable is used in combination with ocean thermal forcing to derive the outlet glacier retreat parameterization. As it is
based on the observed geometry, this is the only variable that does not need to be extended over the tundra.



**Ocean thermal forcing:** We need to know the exact model version of the forcing ESM so we can extract matching ocean data
from the CMIP archive.

Because we calculate anomalies relative to the period 1960-1989, SMB and Tskin have to cover the historical period (1960-
2014) in addition to the projection period (2015-2100). All other data should cover at least the projection period (2015-2100).
In addition, for climate forcing data to be used for ice sheet model initialisation and historical experiments, it should be
provided over the historical period from 1950 under ERA5 forcing or another reanalysis product.
**Appendix B: List of ISM projections**
**Table B1. Ice sheet model versions and number of experiments. Bold model versions are shown in Figure 4.**
Linear - linear sliding law, Weertman - Weertman sliding law (m=1/3), ZI - Zoet-Iverson sliding law, RC - regularised
Coulomb sliding law, PMIP3 - PMIP3 ensemble mean forcing for spinup, HadCM3 - HadCM3 forcing for spinup, CCSM -
CCSM forcing for spinup, MARv3.9 - initialised with MARv3.9, MARv3.12 - initialised with MARv3.12, Num - number of
experiments for different forcings (ESM, scenario, RCM, retreat).

| Group | Model | Resolution (km) | Variant | Num |
|-------|-------|-----------------|---------|-----|
| IGE | ElmerIce2 | 1 - 6 | Linear | 14 |
| | **ElmerIce3** | 1 - 6 | Weertman | 15 |
| IMAU | IMAUICE1 | 40 | ZI, PMIP3 | 57 |
| | IMAUICE2 | 30 | ZI, PMIP3 | 57 |
| | IMAUICE3 | 20 | ZI, PMIP3 | 57 |
| | **IMAUICE5** | 10 | ZI, PMIP3 | 57 |
| | IMAUICE6 | 20 | ZI, HadCM3 | 57 |
| | IMAUICE7 | 20 | ZI, CCSM | 57 |
| | IMAUICE8 | 20 | RC, PMIP3 | 57 |
| NORCE | CISM02-MAR39 | 2 | MARv3.9 | 36 |
| | CISM04-MAR312 | 4 | MARv3.12 | 48 |
| | **CISM04-MAR39** | 4 | MARv3.9 | 69 |
| | CISM04c-MAR39 | 4 | MARv3.9, consistent[†] | 95 |
| | CISM04e-MAR312 | 4 | MARv3.12, extension to 2200 | 5 |
| | CISM04-MAR312 | 4 | MARv3.12 | 4 |
| | CISM08-MAR312 | 8 | MARv3.12 | 48 |
| | CISM08-MAR39 | 8 | MARv3.9 | 115 |
| | CISM08c-MAR39 | 8 | MARv3.9, consistent[†] | 95 |
| | CISM16-MAR312 | 16 | MARv3.12 | 48 |
| | CISM16-MAR39 | 16 | MARv3.9 | 100 |





| | CISM16c-MAR312 | 16 | MARv3.12, consistent[†] | 110 |
| | CISM16oc-MAR39 | 16 | MARv3.9, overshoot to 2300 | 45 |
| | CISM16t-MAR39 | 16 | MARv3.9, repeat to 2300 | 65 |
| | CISM16tc-MAR39 | 16 | MARv3.9, repeat to 2300, consistent[†] | 59 |
| | CISM16xc-MAR12 | 16 | MARv3.12, extension to 2300, consistent[†] | 24 |
| VUB | **GISMHOMv1** | 5 | Higher-order model | 57 |
| | GISMSIAv1 | 5 | Shallow ice approximation | 21 |

[†] retreat sensitivity consistent between historical and projection.

**Appendix C: Construction of extensions until 2300.**

**Extensions under climate forcing IPSL-CM6A-LR SSP5-8.5** have been downscaled with MARv3.13, which is largely similar to v3.12. The only difference is a small correction of albedo in function of the liquid water content of the surface snowpack. Experiment MARv3.13-e05 uses SMB forcing produced at a fixed topography, as for the other experiments. In addition, we have experiment MARv3.13-e55, which uses SMB forcing produced at a changing topography. The topography change was produced by running two iterations between MAR and CISM with consecutive update of SMB and topography. The processing steps were the following:

1. Run MARv3.13 forced with IPSL-CM6A-LR SSP5-8.5 to 2300, where the cumulated SMB anomaly / 4 is used to update the topography. This underestimates the topography change compared a theoretical fully-coupled experiment by around a factor 4, so it is close to no update of the topography.

2. Run CISM with the SMB in 1.

3. Run MARv3.13 forced with IPSL-CM6A-LR SSP5-8.5 to 2300 with topography changes taken every 10 years from 2070 forward from 2.

4. Run CISM with the SMB in 3.

Schematic extension of forcing between 2100 and 2300 based on existing data until 2100.

**Repeat scenarios**. The forcing until 2100 is the same as the corresponding scenario. From 2101 – 2300 the forcing is randomly repeated by shuffling the last 10 years of existing data (2091-2100). The following indices are used.

year = 2101, 2102, 2103, […], 2298, 2299, 2300 ;
shuffled_time_repeat = 2093, 2099, 2095, 2100, 2092, 2097, 2098, 2094,
    2091, 2096, 2100, 2097, 2099, 2095, 2096, 2091, 2093, 2098, 2094, 2092,
    2099, 2095, 2094, 2096, 2091, 2097, 2100, 2093, 2092, 2098, 2100, 2095,
    2098, 2094, 2093, 2097, 2092, 2099, 2096, 2091, 2097, 2098, 2099, 2093,
    2095, 2100, 2092, 2096, 2091, 2094, 2098, 2091, 2094, 2100, 2099, 2092,
    2093, 2096, 2095, 2097, 2100, 2094, 2091, 2096, 2095, 2093, 2092, 2099,
    2097, 2098, 2100, 2098, 2091, 2096, 2093, 2092, 2099, 2094, 2097, 2095,
    2094, 2097, 2095, 2098, 2093, 2092, 2096, 2099, 2100, 2091, 2097, 2095,
    2092, 2094, 2100, 2098, 2099, 2091, 2096, 2093, 2093, 2091, 2096, 2095,





480    2097, 2099, 2098, 2092, 2094, 2100, 2097, 2100, 2098, 2096, 2091, 2094,
481    2099, 2092, 2093, 2095, 2091, 2099, 2100, 2093, 2095, 2094, 2092, 2098,
482    2096, 2097, 2094, 2097, 2095, 2099, 2092, 2098, 2096, 2093, 2100, 2091,
483    2094, 2098, 2093, 2097, 2092, 2100, 2096, 2095, 2091, 2099, 2095, 2091,
484    2096, 2100, 2094, 2097, 2093, 2092, 2098, 2099, 2091, 2094, 2092, 2097,
485    2096, 2100, 2098, 2093, 2099, 2095, 2096, 2091, 2094, 2093, 2098, 2097,
486    2092, 2100, 2095, 2099, 2098, 2091, 2100, 2092, 2097, 2094, 2096, 2093,
487    2099, 2095, 2095, 2096, 2091, 2100, 2099, 2093, 2094, 2092, 2097, 2098;

**Overshoot scenarios**.
Schematic overshoot scenario mimicking SSP5-3.4-OS based on the global temperature evolution. Until year 2055, the forcing
is the same as SSP5-8.5. From year 2056 – 2165, the temperature decreases similarly to the increase between 2030 and 2055
but backwards at 0.25 the rate (drawing four years for one). From 2156 on we shuffle and repeat the forcing earlier in the
experiment, drawn from the time window 2026 – 2038.
Forcing until 2055 is the same as the corresponding SSP5-.8.5 scenario. From 2056 – 2300 the following indices are used.
year = 2056, 2057, 2058, […], 2298, 2299, 2300 ;
shuffled_time_overshoot = 2056, 2056, 2055, 2054, 2053, 2055, 2054, 2053,
498    2052, 2054, 2053, 2052, 2051, 2053, 2052, 2051, 2050, 2052, 2051, 2050,
499    2049, 2051, 2050, 2049, 2048, 2050, 2049, 2048, 2047, 2049, 2048, 2047,
500    2046, 2048, 2047, 2046, 2045, 2047, 2046, 2045, 2044, 2046, 2045, 2044,
501    2043, 2045, 2044, 2043, 2042, 2044, 2043, 2042, 2041, 2043, 2042, 2041,
502    2040, 2042, 2041, 2040, 2039, 2041, 2040, 2039, 2038, 2040, 2039, 2038,
503    2037, 2039, 2038, 2037, 2036, 2038, 2037, 2036, 2035, 2037, 2036, 2035,
504    2034, 2036, 2035, 2034, 2033, 2035, 2034, 2033, 2032, 2034, 2033, 2032,
505    2031, 2033, 2032, 2031, 2030, 2032, 2031, 2030, 2029, 2033, 2038, 2031,
506    2037, 2029, 2035, 2028, 2034, 2036, 2030, 2027, 2032, 2035, 2031, 2037,
507    2029, 2030, 2033, 2032, 2034, 2028, 2038, 2027, 2036, 2032, 2031, 2027,
508    2037, 2034, 2033, 2036, 2035, 2029, 2030, 2038, 2028, 2027, 2034, 2029,
509    2030, 2033, 2036, 2031, 2032, 2035, 2038, 2028, 2037, 2034, 2038, 2027,
510    2036, 2037, 2029, 2035, 2031, 2030, 2032, 2033, 2028, 2036, 2031, 2027,
511    2032, 2030, 2038, 2028, 2034, 2035, 2033, 2029, 2037, 2036, 2028, 2030,
512    2027, 2038, 2032, 2037, 2033, 2034, 2029, 2031, 2035, 2028, 2030, 2035,
513    2033, 2029, 2037, 2034, 2031, 2038, 2032, 2027, 2036, 2032, 2031, 2027,
514    2030, 2028, 2034, 2037, 2035, 2033, 2036, 2038, 2029, 2029, 2036, 2035,
515    2033, 2037, 2028, 2027, 2031, 2038, 2034, 2032, 2030, 2038, 2027, 2033,
516    2037, 2030, 2034, 2036, 2028, 2031, 2029, 2032, 2035, 2038, 2027, 2030,
517    2031, 2034, 2035, 2037, 2036, 2032, 2029, 2033, 2028 ;

**Data Availability**
The forcing data will be provided in ISMIP6 format on a public archive. It consists of SMB and ST anomalies and their
respective vertical gradients that are generic for all ice sheet models. The retreat mask forcing is produced specifically for each
individual ice sheet model version and is maintained by the modellers.



For common analysis, ice sheet model output was conservatively interpolated to a standard 5 km diagnostic grid (same as
ISMIP6). These model output data will be made available on a public archive, while the raw ice sheet model output is the
responsibility of the individual modelling groups.
Projected sea-level contributions will be provided on a public archive.

**Author contributions**
HG designed the experimental setup with input from TE, prepared and distributed the forcing data, collected and processed
the output data, analysed the results and produced the figures. XF, QG, MvdB, BN, RM, MO, FB produced climate forcing
data. HG, CR, CJB, FGC and SL conducted ice sheet model experiments. HG wrote the manuscript with input from all co-
authors.

**Competing interests**
At least one of the (co-)authors is a member of the editorial board of The Cryosphere.

**Disclaimer**
Publisher's note: Copernicus Publications remains neutral with regard to jurisdictional claims made in the text, published maps,
institutional affiliations, or any other geographical representation in this paper. While Copernicus Publications makes every
effort to include appropriate place names, the final responsibility lies with the authors.

**Acknowledgements**
We acknowledge the World Climate Research Programme and its Working Group on Coupled Modelling for coordinating and
promoting CMIP5 and CMIP6. We thank the climate modelling groups for producing and making available their model output
and the Earth System Grid Federation (ESGF) for archiving the CMIP data. We thank the ISMIP6/7 steering committee and
community for defining and providing a framework for this work. Resources were provided by Sigma2 - the National
Infrastructure for High Performance Computing and Data Storage in Norway through projects NN8085K, NN8006K,
NS5011K, NS8006 K and NS8085K. B.N. is a Research Associate of the Fonds de la Recherche Scientifique de Belgique–
F.R.S.-FNRS.

**Financial support**
This research has received funding from the European Union's Horizon 2020 research and innovation programme under grant
agreement 869304 (PROTECT) and has been supported by the Research Council of Norway under project 324639 (GREASE).



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
