# Peer review of "Extending the range and reach of physically-based Greenland ice 1 sheet sea-level projections 2"

_EGUsphere, 2025_

## Author Comment (AC1)

**Authors' reply to reviewer comments**

**Reply to comments by Anonymous Referee #1**

We would like to thank Referee #1 for the very good suggestions and a constructive review. We are confident that we can address the comments (**bold**) in a revised version. Below we provide a point-by-point response to the comments.

Goelzer et al. present the results from a multi-model ensemble of mass-loss/sea-level projections for the Greenland ice sheet, carried out under the PROTECT project. Compared to the Greenland projections for ISMIP6 (Goelzer et al. 2018, 2020), this study includes many improvements, notably (1) the projections begin from simulations that have run through the historical period, (2) global climate model output is downscaled by multiple RCMs, (3) the intermediate 4.5 scenario is included, and (4) some of the runs extends beyond the end of the century to 2300. Although there are fewer participating models than in ISMIP6, the other improvements make this study a valuable step forward. It is likely to be the most comprehensive set of Greenland sea-level projections available between now and when ISMIP7 results are published.

The paper is well organized and clearly written. It presents results in an accessible way without extraneous details. It's possible to read the entire paper and come away with the key messages in an hour or two.

Thank you for the positive evaluation.

My main critique, which will be straightforward to address, is that some of the important results and conclusions do not appear in the Abstract or Section 5. Thus, busy readers (practitioners, for instance) won't find the bottom-line results easily and might be left with questions. For example, the upper end of the 2300 range (3.1 m) is nearly half the total ice-sheet volume and is comparable to the most pessimistic projections for the West Antarctic Ice Sheet. However, this value assumes an extreme emissions scenario that arguably has a very low likelihood. Adding some caveats will reduce the chance that results will be misinterpreted.

We agree that these results should be carefully caveated. The main reason to not include them in the abstract was to avoid them becoming the headline takeaway of the paper without additional context for those reading only the abstract. We will consider this option in a revised version.

**Specific comments**

l. 19: The Abstract does a good job of describing what was done, but it leaves out some key results. For example, one main result is that the projections are very sensitive to the climate scenario (as expected), moderately sensitive to the RCM choice, and relatively insensitive to the ice sheet model choice. The relatively high RCM sensitivity may be surprising for readers not familiar with Glaude et al. (2024). Also, I suggest stating the 2300 high-end

value not only for the extended SSP5-8.5 scenario, but also for the projections with 2100 repeat forcing. Arguably, the repeat-forcing scenario is more realistic and thus more relevant for practitioners than the extended SSP5-8.5 scenario.

It is a good point to include information about the sensitivity to various factors in the abstract to give it more detail. The projections with repeat-forcing are schematic and therefore not more realistic, but we agree that the results are an interesting addition to the abstract.

1. 55: This paragraph describes an important advance on ISMIP6, responding to the criticisms of Aschwanden et al. (2021). Are you able to say (not here, but later in the paper) how much difference this change makes, compared to initializations that do not include a historical simulation?

We have currently no specific experiment that would clearly show the difference, but we suspect that the difference in the projected sea-level contribution is very minor. We will include an estimate or at least a discussion of that point.

1. 95: Please state the equation that describes this parameterization.

OK, will be included.

l. 99: Can you say how the various percentile values of kappa were determined? In particular, how was the 5% value derived? The language suggests that there is a 5% probability that kappa is at least this high, but I suspect there are deep uncertainties here.

This is described in detail in Slater et al. (2020) and illustrated in Figure 5 in Slater et al., (2019). The same principle holds for the 5%. We are not planning to provide these details in the manuscript, but we will guide the interested reader by direct reference to Figure 5 in the 2019 reference.

l. 229: The paper doesn't say how well the 2015 initial ice-sheet geometries compare to observations (e.g., BedMachine ice thickness). I suggest adding a map-view figure showing (1) the observed ice thickness from BedMachine and (2) the thickness error at 2015 for one ensemble member for each ISM, perhaps indicating on each panel the rms thickness error over the ice-covered area.

OK, we will include such a figure.

Optionally, you could also add a graph or table comparing (for each ISM) the simulated mass change over the historical period to the estimated observed mass loss.

We will consider that option.

l. 236: The text cites O'Neill et al. (2021) for the 2300 scenario. This should be O'Neill et al. (2016). I suggest stating that the maximum CO2 concentration for this scenario is about 2200 ppm (see Fig. 5b in that paper), roughly double the value at 2100. This explains the

large differences in 2300 between the repeat-forcing experiments and those based on ScenarioMIP.

Thanks for spotting that. And for the suggestions, which will be a useful addition.

l. 255: "another problem on this timescale may be that the climate response to changing ice sheet geometry is not properly accounted for". Isn't this also true for the "natural extensions"? The IPSL and CESM2 extensions weren't run with interactive ice sheets, so in neither case would the ESM simulate the climate response to changing ice sheet geometry. Or is there a subtlety I'm missing?

That's correct, well spotted. We will reformulate that to apply to all extensions.

I want to suggest that the issues with schematic repeat forcing are not necessarily worse than those associated with CO2 of 2200 ppm. Neither scenario is fully realistic, but the two approaches are complementary and are probably the best we can do with the CMIP6 output we have.

We appreciate that suggestion and would add that "realistic" is a difficult term when talking about events more than 250 years in the future. However, considerable effort went into the construction of the ScenarioMIP extensions, and they are consistent in that sense, which one cannot say about the repeat forcing.

l. 323: Can you say why the SMB in RACMO is less negative for future projections than those in MAR and HIRHAM? This is discussed by Glaude et al. (2024), but it would be helpful to give a summary here.

OK, we will add a brief summary.

Figure 6:

Please state in the caption that this is for 2100.

OK, will be added

In panel c, I suggest showing the ranges separately for 8.5 repeat forcing and 8.5 extended forcing, since the former is arguably more plausible than the latter.

The extensions start after 2100 and are not relevant here.

It would be helpful to use a different symbol for the CISM runs with extreme values of kappa – e.g., an open green circle instead of a closed green circle. This would confirm that the range of CISM results is close to that of the other ISMs when large kappa is excluded.

We have resisted to add more differentiation to these figures, but this may be one that is quite useful. We will try.

Figure 8: I found this figure confusing because there are experiments of the same type on both sides of the dashed lines, but it's hard to compare them due to the different vertical scales. I suggest using a different color for each type of forcing: o2300, r2300, x2300, and e2200. Another possibility might be having a separate panel for each type (maybe combining e2200 with x2300), with a different vertical scale for the natural extensions.

For guidance: on the left are the overshoot runs and scenarios SSS126 and SSP245. On the right are only SSP585 experiments. In our logic the color is reserved, because these are all CISM experiments.

l. 371: Please state the ranges for Goelzer et al. (2020) and Payne et al. (2021). It might be worth noting that the Payne et al. ranges are much larger than those in Goelzer et al., and similar to the ranges in this study, because of the high warming in some CMIP6 models compared to CMIP5.

Good point, we will add the ranges.

1. 374: "A solid number" is vague; can you make this more quantitative?

Sure, a concrete number will be added here.

Somewhere in the Discussion, I suggest commenting on the value of a "mini-MIP" like this one. Does it save time, without diminishing the results, to run just four models instead of a larger number? Other MIP groups might see this study as a template for efficiently updating projections between IPCC reports when resources are limited.

Good suggestions. We will comment on that. A "mini-MIP" is definitely easier to manage, but no replacement for engaging the full community, not only because of the results.

I also suggest stating that the lack of climate feedbacks (as would be present in an ESM with interactive ice sheets) is an important limitation of the study.

Will be added.

l. 411: Since some readers will skim the Abstract and then go straight to the Conclusions, I suggest adding a paragraph summarizing the main results. I think some redundancy with the Discussion is okay. For example:

Give the total GMSL ranges for the different scenarios and time frames. Restate the result that under high forcing, the ESM choice is the largest contributor to the uncertainty, followed by RCM choice and ISM choice. The rather large sensitivity to RCM choice points to the need for better understanding of the downscaling. You might add some thoughts on how ISMIP7 will build on this study.

OK, we will add a paragraph summarizing the main results.

**Minor fixes**

All minor corrections will be addressed.

**Reply to comments by Anonymous Referee #2**

We would like to thank Referee #2 for the thorough and valuable comments. We appreciate the opportunity to address the comments (**bold**) in a revised version. Below we provide a point-by-point response to the comments.

The authors present results from a multi-model ensemble of future Greenland ice sheet projections carried out within the PROTECT project. This study includes several improvements compared to the ISMIP6 simulations, i.e., it includes different regional climate models for the downscaling (compared to just one in ISMIP6), the protocol has been extended over the historical period, and intermediate CO2 scenario runs have been included. The authors present ranges of potential sea level contributions with respect to the scenarios, regional climate models, and ice sheet models used. This study clearly adds nicely to the ISMIP6 results and comprises valuable results for assessing the potential behavior of the Greenland ice sheet under future warming and the uncertainties associated with it.

The paper is well structured and well written. The figures are of high quality and informative.

Thank you very much for the positive feedback.

**General comments:**

I think it would be helpful for the reader to include additional figures regarding the initial ice sheet state and the climate forcing used.

Time series of the global mean temperature would allow for assessing the effect of the ESM's climate sensitivity on the ice sheet response.

Except for the construction of the overshoot scenarios, global mean temperature is not part of the model forcing. But we agree that it could be useful to provide it to give context for the projections. This will be added.

As one huge advantage of this study compared to ISMIP is the historical simulations, I think it would be helpful to include time series of the SMB for different models and/or a map of the initial ice sheet showing how well current observations of the ice sheet extent are captured by the ensemble.

We will add a figure of the initial states for selected ice sheet models in line with a similar request of Referee #1.

The authors state the importance of emulators and that their model results are geared towards emulator use. However, I miss some more explanation of this. As I understand, the ensemble has been updated with additional experiments to meet the needs of emulators. Except for this, were there additional requirements for the model output to be met in order to facilitate the use of emulators?

No. The targeted emulator had already used output from ISMIP6 and no additional requirements were needed. It ingests time series of the ice sheet sea-level contribution, a standard output from the ice sheet models.

**Specific comments:**

L28f: Given that a large share of the spread is due to different scenarios, it might be more informative to give the GMSL spread here scenario-specific.

Agreed, this will be a good addition.

L95: Please consider showing the function for the parametrization here.

The actual function is

$$L = K \Delta(Q^{0.4} TF)$$

where Q is the mean summer (June–July–August) subglacial runoff (in  $m^3$  s-1) and TF is the ocean thermal forcing (in  ${}^{\circ}$ C).

But we are not convinced adding that much detail is useful. Instead, we have described the main dependencies and refer to the original publications.

L99: After reading Slater et al. (2019), it became clear to me what you mean by "can be expressed probabilistically and is sampled ...". However, I think it would be helpful to include the PDF here and indicate which values you take for your simulations.

Like for the last point, we feel this is adding too much detail. Instead, we will refer specifically to Figure 5a in Slater et al., (2019) to guide the interested reader better.

L125: "GIMP DEM": not mentioned earlier. Please explain the acronym.

GIMP: Greenland Ice Mapping Project. Will be added.

Table 2: Please avoid the page break in the table.

Will be taken care of in the final formatting of the paper.

L134: Should be Appendix A, I assume.

Yes, will be corrected.

L247: Should be Appendix C, I assume.

Yes, will be corrected.

L264: Should be Appendix C, I assume. Here, I hoped to see some time series of the forcing data, e.g., global mean air temperature.

Yes, that was the intention. Will be added.

Figure 5: Given the large spread of the CISM results, would it be possible to use different markers for the different retreat parametrizations?

The information density of this figure is probably already at its limit, but we will try to include that, in line with a similar request of Referee #1 for Figure 6.

L366: Should be "rightmost" instead of "leftmost"

Thanks for spotting that.

---

## Author Response (AR1)

**Authors' reply to reviewer comments**

**Reply to comments by Anonymous Referee #1**

We would like to thank Referee #1 for the very good suggestions and a constructive review. We are confident that we have addressed the comments (**bold**) in the revised version. Below we provide a point-by-point response to the comments.

**Goelzer et al. present the results from a multi-model ensemble of mass-loss/sea-level projections for the Greenland ice sheet, carried out under the PROTECT project. Compared to the Greenland projections for ISMIP6 (Goelzer et al. 2018, 2020), this study includes many improvements, notably (1) the projections begin from simulations that have run through the historical period, (2) global climate model output is downscaled by multiple RCMs, (3) the intermediate 4.5 scenario is included, and (4) some of the runs extends beyond the end of the century to 2300. Although there are fewer participating models than in ISMIP6, the other improvements make this study a valuable step forward. It is likely to be the most comprehensive set of Greenland sea-level projections available between now and when ISMIP7 results are published.**

**The paper is well organized and clearly written. It presents results in an accessible way without extraneous details. It's possible to read the entire paper and come away with the key messages in an hour or two.**

Thank you for the positive evaluation.

**My main critique, which will be straightforward to address, is that some of the important results and conclusions do not appear in the Abstract or Section 5. Thus, busy readers (practitioners, for instance) won't find the bottom-line results easily and might be left with questions. For example, the upper end of the 2300 range (3.1 m) is nearly half the total ice-sheet volume and is comparable to the most pessimistic projections for the West Antarctic Ice Sheet. However, this value assumes an extreme emissions scenario that arguably has a very low likelihood. Adding some caveats will reduce the chance that results will be misinterpreted.**

We agree that these results have to be carefully caveated and have done so in the revised version. The main results of different extensions to 2300 are now included in the abstract. The results section has been modified accordingly to bring out these numbers and frame them better.

**Specific comments**

**l. 19: The Abstract does a good job of describing what was done, but it leaves out some key results. For example, one main result is that the projections are very sensitive to the climate scenario (as expected), moderately sensitive to the RCM choice, and relatively insensitive to the ice sheet model choice. The relatively high RCM sensitivity may be surprising for readers not familiar with Glaude et al. (2024). Also, I suggest stating the 2300 high-end**

**value not only for the extended SSP5-8.5 scenario, but also for the projections with 2100 repeat forcing. Arguably, the repeat-forcing scenario is more realistic and thus more relevant for practitioners than the extended SSP5-8.5 scenario.**

We have included more information about the sensitivity to various factors in the abstract and also results for the projections with repeated forcing.

**l. 55: This paragraph describes an important advance on ISMIP6, responding to the criticisms of Aschwanden et al. (2021). Are you able to say (not here, but later in the paper) how much difference this change makes, compared to initializations that do not include a historical simulation?**

We have currently no specific experiment that would clearly show the difference, but results by Rahlves et al. (2025) give indications that the effect on the projected sea-level contribution is minor. This has been added in the discussion.

**l. 95: Please state the equation that describes this parameterization.**

The actual function for the retreat L is
$$L = K \Delta(Q^{0.4}\,TF)$$
where Q is the mean summer (June–July–August) subglacial runoff (in $m^3\,s^{-1}$), TF is the ocean thermal forcing (in ∘C) and $K$ is the retreat parameter.
But we are not convinced adding that much detail is useful. Instead, we have described the main dependencies and refer to the original publications.

**l. 99: Can you say how the various percentile values of kappa were determined? In particular, how was the 5% value derived? The language suggests that there is a 5% probability that kappa is at least this high, but I suspect there are deep uncertainties here.**

This is described in detail in Slater et al. (2020) and illustrated in Figure 5 in Slater et al., (2019). The same principle holds for the 5%. We do not want to overload the manuscript with these details and instead guide the interested reader by direct reference to Figure 5 in the 2019 reference.

**l. 229: The paper doesn't say how well the 2015 initial ice-sheet geometries compare to observations (e.g., BedMachine ice thickness). I suggest adding a map-view figure showing (1) the observed ice thickness from BedMachine and (2) the thickness error at 2015 for one ensemble member for each ISM, perhaps indicating on each panel the rms thickness error over the ice-covered area.**

Thank you for the suggestion, we have included such a figure in the document.

**Optionally, you could also add a graph or table comparing (for each ISM) the simulated mass change over the historical period to the estimated observed mass loss.**

We have considered this option but decided against it to limit the amount of figures, which is already quite large.

**l. 236: The text cites O'Neill et al. (2021) for the 2300 scenario. This should be O'Neill et al. (2016). I suggest stating that the maximum CO2 concentration for this scenario is about 2200 ppm (see Fig. 5b in that paper), roughly double the value at 2100. This explains the large differences in 2300 between the repeat-forcing experiments and those based on ScenarioMIP.**

Thanks for spotting that and for the suggestion. We have included information about the extensions in the revised text.

**l. 255: "another problem on this timescale may be that the climate response to changing ice sheet geometry is not properly accounted for". Isn't this also true for the "natural extensions"? The IPSL and CESM2 extensions weren't run with interactive ice sheets, so in neither case would the ESM simulate the climate response to changing ice sheet geometry. Or is there a subtlety I'm missing?**

That's correct, well spotted. We have moved the text further below and reformulated to apply to all extensions.

**I want to suggest that the issues with schematic repeat forcing are not necessarily worse than those associated with CO2 of 2200 ppm. Neither scenario is fully realistic, but the two approaches are complementary and are probably the best we can do with the CMIP6 output we have.**

We appreciate that point of view and would add that "realistic" is a difficult term when talking about events more than 250 years in the future. However, considerable effort went into the construction of the ScenarioMIP extensions, and they are consistent in that sense, which one cannot say about the repeat forcing.

**l. 323: Can you say why the SMB in RACMO is less negative for future projections than those in MAR and HIRHAM? This is discussed by Glaude et al. (2024), but it would be helpful to give a summary here.**

OK, we have added a brief summary.

**Figure 6:**

**Please state in the caption that this is for 2100.**

OK, has been added

**In panel c, I suggest showing the ranges separately for 8.5 repeat forcing and 8.5 extended forcing, since the former is arguably more plausible than the latter.**

The extensions start after 2100 and are not relevant here. See your last point.

**It would be helpful to use a different symbol for the CISM runs with extreme values of kappa – e.g., an open green circle instead of a closed green circle. This would confirm that the range of CISM results is close to that of the other ISMs when large kappa is excluded.**

We have now plotted results for the extreme kappa values in figure 6, separated in panel a, and then overlapping in the remaining panels. We consider this a good compromise as it makes it possible to compare the ranges with the other ISMs.

**Figure 8: I found this figure confusing because there are experiments of the same type on both sides of the dashed lines, but it's hard to compare them due to the different vertical scales. I suggest using a different color for each type of forcing: o2300, r2300, x2300, and e2200. Another possibility might be having a separate panel for each type (maybe combining e2200 with x2300), with a different vertical scale for the natural extensions.**

For guidance: on the left are the overshoot runs and scenarios SSP1-2.6 and SSP2-4.5. On the right are only SSP5-8.5/RCP8.5 experiments. We have updated the caption accordingly. In our logic the colour is reserved, because these are all CISM experiments.

**l. 371: Please state the ranges for Goelzer et al. (2020) and Payne et al. (2021). It might be worth noting that the Payne et al. ranges are much larger than those in Goelzer et al., and similar to the ranges in this study, because of the high warming in some CMIP6 models compared to CMIP5.**

Good point. We have added the combined (CMIP5 and CMIP6) range from Payne et al., 2021, as this makes for a more meaningful comparison to our mixed CMIP5 and CMIP6 ensemble. Differences between CMIP5 and CMIP6 are discussed elsewhere.

**l. 374: "A solid number" is vague; can you make this more quantitative?**

OK, we have added the actual number of experiments, which is 159.

**Somewhere in the Discussion, I suggest commenting on the value of a "mini-MIP" like this one. Does it save time, without diminishing the results, to run just four models instead of a larger number? Other MIP groups might see this study as a template for efficiently updating projections between IPCC reports when resources are limited.**

Good suggestions. We have commented on that in the new version. A "mini-MIP" is definitely easier to manage, but no replacement for engaging the full community, not only because of the results.

**I also suggest stating that the lack of climate feedbacks (as would be present in an ESM with interactive ice sheets) is an important limitation of the study.**

Thanks. Has been added.

**l. 411: Since some readers will skim the Abstract and then go straight to the Conclusions, I suggest adding a paragraph summarizing the main results. I think some redundancy with the Discussion is okay. For example:**

**Give the total GMSL ranges for the different scenarios and time frames.**
**Restate the result that under high forcing, the ESM choice is the largest contributor to the uncertainty, followed by RCM choice and ISM choice. The rather large sensitivity to RCM choice points to the need for better understanding of the downscaling.**
**You might add some thoughts on how ISMIP7 will build on this study.**

We have specified additional sea-level ranges in the main text, in the discussion and in the abstract and added some top level results in the conclusion.

**Minor fixes**

**Please make sure there is a carriage return between each paragraph, or some other indication of a paragraph break.**
This is taken care of during typesetting of the manuscript.

**For figures, put just the label (e.g., Figure 1) in boldface, instead of the entire caption.**
OK

**l. 87 I think it's more common to denote surface temperature by 'TS' than 'ST'.**
We keep ST for consistency with the provided forcing datasets.

**Table 1: Change "RCP5.8" to "RCP8.5" in the header.**
OK

**l. 126: No hyphen in "grid cells"**
OK

**l. 162: CO2 -> $CO_2$ (also in l. 164)**
OK

**l. 174: The units here are not formatted correctly.**
OK

**l. 350: 6% (no space)**
OK

**l. 356: Add a comma after "scenario SSP1-2.6" for clarity.**
OK

**l. 423: I think a word is missing after "text".**
OK

**l. 427:  "heigh" -> "height"**
OK

**l. 454:  "in function" -> "as a function"?**
OK

**l. 459:  The "/ 4" means that a quarter of the anomaly was applied? Maybe change the wording for clarity.**
OK

**l. 460:  "compared" -> "compared to"**
OK

**l. 491: This is a sentence fragment; it needs a verb.**
OK

**l. 495: Extraneous decimal point in "SSP5-.8.5"**
OK

**Reply to comments by Anonymous Referee #2**

We would like to thank Referee #2 for the thorough and valuable comments. We appreciate the opportunity to address the comments (**bold**) in a revised version. Below we provide a point-by-point response to the comments.

**The authors present results from a multi-model ensemble of future Greenland ice sheet projections carried out within the PROTECT project. This study includes several improvements compared to the ISMIP6 simulations, i.e., it includes different regional climate models for the downscaling (compared to just one in ISMIP6), the protocol has been extended over the historical period, and intermediate CO2 scenario runs have been included. The authors present ranges of potential sea level contributions with respect to the scenarios, regional climate models, and ice sheet models used. This study clearly adds nicely to the ISMIP6 results and comprises valuable results for assessing the potential behavior of the Greenland ice sheet under future warming and the uncertainties associated with it.**

**The paper is well structured and well written. The figures are of high quality and informative.**

Thank you very much for the positive feedback.

**General comments:**

**I think it would be helpful for the reader to include additional figures regarding the initial ice sheet state and the climate forcing used.**
**Time series of the global mean temperature would allow for assessing the effect of the ESM's climate sensitivity on the ice sheet response.**

Just to clarify, except for the construction of the overshoot scenarios, global mean temperature is not part of the model forcing. But we agree that it is a useful diagnostic to provide context for the projections, especially the extreme cases. We have added a panel with global mean temperature evolution for the natural extensions to 2300 and included a figure for all experiments to 2100 in the supplement.

**As one huge advantage of this study compared to ISMIP is the historical simulations, I think it would be helpful to include time series of the SMB for different models and/or a map of the initial ice sheet showing how well current observations of the ice sheet extent are captured by the ensemble.**

We have added a figure of the initial states for selected ice sheet models in line with a similar request of Referee #1.

**The authors state the importance of emulators and that their model results are geared towards emulator use. However, I miss some more explanation of this. As I understand, the ensemble has been updated with additional experiments to meet the needs of emulators.**

**Except for this, were there additional requirements for the model output to be met in order to facilitate the use of emulators?**

No. The targeted emulator had already used output from ISMIP6 and no additional requirements were needed. It ingests time series of the ice sheet sea-level contribution, a standard output from the ice sheet models.

**Specific comments:**

**L28f: Given that a large share of the spread is due to different scenarios, it might be more informative to give the GMSL spread here scenario-specific.**

Agreed, scenario ranges have been added. We have also corrected a rounding error for the upper bound, which is 354 not 353 mm.

**L95: Please consider showing the function for the parametrization here.**

See response to similar comment from Referee #1

**L99: After reading Slater et al. (2019), it became clear to me what you mean by "can be expressed probabilistically and is sampled …". However, I think it would be helpful to include the PDF here and indicate which values you take for your simulations.**

Like for the last point, we feel this is adding too much detail. Instead, we refer specifically to Figure 5a in Slater et al., (2019) to guide the interested reader better.

**L125: "GIMP DEM": not mentioned earlier. Please explain the acronym.**
GIMP: Greenland Ice Mapping Project. Has been added.

**Table 2: Please avoid the page break in the table.**
This is taken care of in the final formatting of the paper.

**L134: Should be Appendix A, I assume.**
Yes, has been corrected.

**L247: Should be Appendix C, I assume.**
Yes, has been corrected.

**L264: Should be Appendix C, I assume. Here, I hoped to see some time series of the forcing data, e.g., global mean air temperature.**

Yes, correct.
We have added timeseries plots of global mean air temperature in the supplement.

**Figure 5: Given the large spread of the CISM results, would it be possible to use different markers for the different retreat parametrizations?**

We have used different markers, in line with a similar request of Referee #1 for Figure 6.

**L366: Should be "rightmost" instead of "leftmost"**
Thanks for spotting that. Corrected.

---

## Author Response (AR2)

**Authors' reply to editor's comments**

We would like to thank the Editor for the detailed comments. We are confident that we have addressed the comments (**bold**) in the revised version. Below we provide a point-by-point response.

**Overall, the manuscript is in good shape and has addressed the comments of the reviewers well. Nonetheless, there are several minor improvements to be made. Two relatively general comments are given here:**

**- Section 2.6 and Appendix A are both a bit strangely introduced in this text, and I think Section 2.6 doesn't belong where it is. Mainly the phrases "Data request for ice sheet model output" and "The requested data" are a bit strange, as this section is intended to describe the data that is available and has been produced in this study right? There has not been a request from someone yet, or the request already happened internally within the group of contributors. Maybe reformulate to "Data availability" and "Data from this study consists of ...". More broadly, I would change current Appendix A to A.1, and move Section 2.6 to Appendix A.2. Reformulate without the phrase "request" anywhere, or make it clear that this relates to the internal requests within your consortium, it is not a current request to the community.**
We have integrated information from 2.6 into section 'Data Availability' and reformulated other passages mentioning "data request". We have also revised Appendix A to make it clearer.

**- Second, in regards to possible future work with this ensemble using an emulator, in this work I think it is appropriate to avoid referring to a concrete emulator or study, as none was described here. There are a few comments along these lines below.**
OK

**Along with these general comments, please find additional specific comments for minor revisions below. I look forward to a revised version of the manuscript.**

**Specific comments**

**L19-23: These first three sentences are a bit redundant and confusing. Is the study presenting the projections or making use of them? I recommend revising for clarity here.**
OK, shortened and reformulated.

**L22: The focus is on providing → Here we provide**
OK

**L25: Consider deleting "some of the"**
OK

**L53: ice sheet scale → ice-sheet scale**
It seems both spellings are possible. We prefer the version without hyphen, consistent with our preferred spelling of "ice sheet model", "ice sheet geometry", …

**L56: ice sheet projections → ice-sheet projections [etc, check throughout]**
Same as above

**L69: it requires statistical tools → statistical tools are needed**
OK

**L96 (equation): I am a strong proponent of following mathematical notation conventions in equations, which means e.g. 1 letter per variable, subscripts, etc. I would suggest reformulating this equation in this way. For example, I believe $a(x,y,t) = a_\mathrm{ref}(x,y)+a_\mathrm{anom}(x,y,t)+\frac{da}{dz}(x,y,t) \cdot \Delta z(t)$ would a clearer way to write this equation. Greve and Blatter (2009) use "a" for the accumulation-ablation term (ie, SMB), or Cuffey and Patterson (2010) use "b" for balance. I realize you may want to keep consistency with e.g. ISMIP6 and past work, so I would not consider a change mandatory. But I think it would improve readability here and then throughout the text.**
As suspected, we would like to keep consistency with ISMIP6 and other recent work and have kept the less formal equation.

**L106: kappa → κ**
OK

**L145: in familiar format → in a familiar format**
OK

**L245-246: "one statistical downscaling method" ← Earlier it was stated that this would be considered as an RCM (L117-118). Make sure to frame this consistently throughout the paper. Perhaps the statement can be rephrased on L117-118 to simply say it can be considered similar in capability to an RCM in terms of the forcing it provides, if you want to continue to refer to it separately.**
We have used the suggested formulation in L117-118 and adapted the text elsewhere. We believe it is clear from that point on that descriptions stating 'RCMs' include the statistical downscaling approach even if it is not a formal definition.

**L256: to continue downscaling → of further downscaling of [or] of continuing to downscale**
OK

**L287: from the beginning a modelling strategy → a modelling strategy from the beginning**
OK

**L302: choices to cover → choices, intended to cover**
OK

**FIgure 6: There is an inconsistency between Figs. 5 & 6, namely that in Fig. 5 it is clear that the VUB-GISM simulations gives higher and indeed the maximum SLCs by 2100, but in Fig. 6, NORCE-CISM is systematically higher and represents the maximum. I understand**

**this is because Fig. 5 shows a particular subset. Would it be possible to add something to Fig. 6, to identify those simulations that were plotted in Fig. 5? Perhaps a vertical grey line behind each symbol that is associated with Fig. 5?**
We believe this comment is similar to an issue raised earlier by a reviewer for what is now figure 7. We have addressed it in a similar way by distinguishing symbols for the extreme kappa settings in figure 6. We have also modified the caption for figure 5 to more clearly indicate that it illustrates a subset of experiments.

**L334-335: I would suggest listing the (a), (b), ... before the item, so (a) ISMs, (b) RCMs, (c) scenarios and (d) CMIP iterations. This is more usual and eases readability. Or better, remove, the (a), (b), ... entirely, since this really about the figure, right? This ordering information is contained within the caption of the Figure itself.**
OK, removed.

**L371: climate forcing. → climate forcing,**
OK

**Figure 9: This figure should be improved significantly. Currently it relies heavily on the long programmatic x-labels to inform the reader of what they are looking at. First I would suggest, keeping one shared y-axis throughout. But rather have the lower 1/3 of the axis spaced evenly from 0-250 mm, then the upper 2/3 of the axis could show values from 250-3000mm. Then I would include some kind of light shading grouping the experiments together, as far as I can see, this is first by scenario, so groups of [ssp585-o2300, ssp126-r2300, ssp245-r2300, rcp85/ssp585-r2300, ssp585-x2300, ssp585-e220], with the latter group set to one side since it is further distinguished from the others. The shaded bands could have text labels directly or a corresponding legend. Then the x-labels could simply be comprised of the climate model and the choice of regional downscaling. This would be one solution to improve this figure.**
Thanks for the suggestions. We have updated the figure largely along those lines. Instead of different vertical scaling in the lower and upper range, we have opted for a logarithmic scale throughout.

**L398: less ISMs → using a subset of ISMs**
OK

**L398: due to additional → due to the incorporation of additional**
OK

**L402: the subsequent → possible future**
OK

**L415: is dominating → dominates**
OK

**L419: we have explored here in addition → we have additionally explore here**
OK

**L430: source of uncertainty → source of uncertainty on this timescale**
OK

**L432: going to 2300 → going to 2300 or even beyond**
OK

**L437-439: This sentence needs to be revised. Consider reordering the sentence, something like "Combined, these results indicate that modifications to the ISMIP6 forcing protocols and new methods to account for a changing ice sheet geometry are needed for robust standalone ice sheet simulations well beyond year 2100".**
OK

**L443: provide the emulator → provide an emulator**
OK